# Predictors of Viewing YouTube Videos on Incheon Chinatown Tourism in South Korea: Engagement and Network Structure Factors

Woohyun Yoo 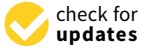, Taemin Kim and Soobum Lee *

Department of Mass Communication & Institute of Social Sciences, Incheon National University, Incheon 22012, Korea; woohyunyoo@gmail.com (W.Y.); taemin.kim@inu.ac.kr (T.K.)
* Correspondence: soolee@inu.ac.kr

**Abstract:** YouTube has become an increasingly popular source of tourism information. The purpose of this study is to explore the network structures of YouTube videos about Incheon's Chinatown in South Korea and investigate the potential factors that can predict the viewing of these videos. The analysis of 104 videos about Incheon Chinatown revealed that the engagement factors assessed by the number of comments and likes, and the running time of content, were significant predictors of viewing. However, network structure factors did not predict viewing. These findings make valuable contributions to sustainable tourism research and provide practical guidance for tourism management.

**Keywords:** sustainable tourism; tourism; social media; YouTube; social network analysis

## 1. Introduction

Incheon is one of South Korea's typical cities, with various tourism resources and a convenient transportation infrastructure is centered on the Incheon International Airport. Chinatown is a representative historical and cultural tourist attraction in Incheon and is visited by numerous foreign tourists each year [1]. Chinatown has many historical local restaurants run by local people, and it serves as a gateway to some eco-friendly tourist spots, such as tidal flats and islands associated with Incheon. Furthermore, Incheon Chinatown was the first place where foreign civilizations and cultures were introduced to South Korea and made a significant contribution to the growth of the local economy in Incheon. However, only 8.3% of the foreign tourists who come to South Korea currently visit Incheon [2]. Thus, it is important to promote Incheon Chinatown to increase its popularity among foreign tourists and to develop the local economy by reviving Incheon Chinatown.

Social media has become an increasingly popular source of tourism information [3]. In particular, YouTube has attracted attention as a medium for tourism promotions. In 2014, the number of subscribers for the top travel channels on YouTube increased by about 106% from the previous year. Among these, the number of subscribers to video blogs (vlogs) that feature personal travel experiences accounted for 48% of this total increase [4]. This finding implies that the sharing of individual experiences on YouTube plays an influential role in tourism promotion.

The sharing and disseminating of travel-related information on YouTube can be explained by the electronic word of mouth (e-WOM) phenomenon. Tourists can exchange travel-related information, personal experience, and thoughts via text comments, photos, and videos on YouTube both during and after traveling [3,5]. This implies that a new paradigm for spreading tourism relevant information is being developed. In other words, it has become important to identify how e-WOM functions in order to maximize its effects [6]. To develop effective sustainable tourism strategies and to deal with the growth of visitors at destinations, it is critical to investigate the major factors that affect the formation of a

network of numerous relationships, and examine how this spreads. Although researchers have conducted studies to understand predictors of the popularity of YouTube videos about a variety of topics, such as politics, health, education, and science [7–10], little attention has been paid to the video content on YouTube about Incheon Chinatown and how the content disseminates to users and potential inbound travelers to South Korea.

Thus, the current study aimed to explore the spreading of the YouTube content about Incheon Chinatown, and investigate the potential factors that can promote the viewing of this content. To do so, we collected the video clips from YouTube by using the relevant search words on Incheon Chinatown. Then, a network analysis was conducted to investigate the network structure of those videos and what factors predicted the popularity of the videos through NodeXL.

## 2. Literature Review

### 2.1. Heritage Tourism and Chinatown in Incheon

Heritage tourism refers to a type of tourism focused on the commercialization of historical and sociocultural aspects to draw tourists [11]. The local traditions, such as arts crafts, cuisine, and dances, as well as the built heritage, including cathedrals, factories, museums, and markets, function as main assets in the heritage tourism industry [12]. Heritage tourism is also regarded as means of economic development that achieves growth through attracting foreign tourists [13]. From a heritage tourism perspective, Chinatowns are important heritage spaces that attract and educate tourists. Chinatowns were developed as ethnic enclaves that reflected the socio-economic segregation of Chinese immigrants in a hostile host country. Over time, Chinatowns have evolved, and their improved integration into the broader society, increased emphasis on multiculturalism, and interest in leveraging Chinatown as a heritage space for tourism [14].

Incheon Chinatown is the only place in South Korea where the original shape of the Chinese street is still preserved, and it is a representative historical and cultural attraction in Incheon. However, due to recent political and diplomatic conflicts with China, such as the deployment of the Terminal High Altitude Area Defense (THAAD), the Korean wave has been virtually banned in China, which is a major target country of the South Korean tourism industry. The number of Chinese tourists, accounting for 47.3% of the foreign tourists in July 2016, reached a record high of 920,000 [2]. However, after the THAAD deployment decision, this number has continuously dropped, decreasing sharply to about 360,000 in March 2017 [2]. Thus, the overall tourism industry forecast is currently unclear. Incheon's Chinatown, which attracts the highest number of Chinese tourists, is no exception. With the damage to the tourism industry caused by the THAAD deployment, the limitation associated with excessive dependency on China was revealed through a practical indicator. Consequently, there have been suggestions that the tourism target countries need to be diversified in order to respond flexibly to rapidly changing world situations. Likewise, because Incheon has a high dependency on Chinese tourists, it is now necessary to develop influential factors that can attract tourists from various countries for the continuous progress of Incheon's culture and tourism.

### 2.2. Online Travel Information, Social Media, and One-Person Media

The advancement of Internet technology has changed the structure of the tourism industry. In the current situation, where the paradigm of tourism is changing from quantity to quality, both tourism consumers and suppliers exchange tourism information using various platforms on the Internet [15]. Thus, local governments bolster their online promotional activities utilizing their resources on online video channels and tourism markets. Travelers also collect various information related to tourism destinations or products by actively using the Internet to gather information and reduce the uncertainties as much as possible.

Among the several types of information and communication technologies (ICT), social media play an increasingly important role in tourism, particularly for travel-related

information searching and decision-making processes. For example, consumers obtain information to assist in a trip planning process and to make informed decisions about destinations, accommodation, restaurants, tours, and attractions [3,16]. Travel companies also develop their social media to engage potential travelers in conversations. However, these changes present new challenges as well as opportunities for tourism destinations and suppliers. People are becoming more difficult for companies and marketers to reach due to the number of advertising messages they encounter day after day. In this situation, social media serves as a great marketing channel through which destinations and tourism enterprises can reach and persuade potential visitors [17].

In addition, social media blur the boundaries between producers and consumers and enable all participants to be users as well as producers of information and knowledge. With the advent of social media, everyone can serve as a creator and distributor of content. Tourists are able to share travel-related information, personal experiences, and thoughts via text comments, photographs, and videos on social media [3,5]. The advent of one-person-media seems to have accelerated this change and encouraged local governments and travelers to be more active online. One-person-media refers to a new social media platform that involves a one-person creator who produces and shares content related to various subjects by themselves. Currently, travelers use one-person media to promote tourism products, or to collect tourism information. Furthermore, due to the advancement of Internet technology and the widespread use of smartphones, it is now possible to use and operate one-person-media anytime and anywhere, increasing its influence even further.

In South Korea, as one-person broadcasts have become popular and its potential has been recognized, various one-person broadcasts are now produced and distributed on YouTube. The influential one-person creator-oriented broadcasts on domestic tourism products consist of the following basic content: he/she visits a famous restaurant in a certain region and after tasting the food, he/she evaluates the taste and provides a review; or he/she goes to a famous tourism destination and introduces relevant tourism products as a reporter. "*Korean Englishman*" is a typical one-person broadcast on South Korean tourism and culture. This broadcast channel has about 1.9 million subscribers and covers South Korean food, culture, and travel, accompanied by foreign acquaintances who are visiting South Korea for the first time. Furthermore, the one-person broadcast channels on tourism and travel are continuously posting various content to provide advice for people visiting South Korea by recommending places to visit when traveling to a certain region.

*2.3. Determinants of Viewing Tourism Videos on YouTube: Engagement and Network Structure Factors*

While planning a trip, searching for information on tourist destinations has a large effect on the tourism experience, as well as on overall satisfaction. From the marketing perspective, the expectations during an activity is an important factor that can influence its satisfaction [18]. The recent rapid advancement of information and communication technologies and the advent of the web 2.0 era, which implements the values of openness, participation, and sharing, have led to many changes in all the processes of information searching, tour planning, and the final experience of tourists [19]. In particular, the advent of social network sites (SNSs) allows users to inexpensively and reliably share information anytime and anywhere; thus, it is used as a major tool for delivering and acquiring information about tourism destinations and products. Furthermore, the delivery of travel information and exposure to tour destination images using video portals such as YouTube help form positive images for potential tourists, thereby having a positive effect on tourism decision making.

The popularity of YouTube in tourism marketing has encouraged scholarly interests regarding the predictive indicators of YouTube content consumption. Some scholars have focused on the notion of user engagement that includes all forms of user attention and involvement with social media [20,21]. This engagement is often viewed as interactivity, which enables users to not only receive information but also send and disseminate it. On YouTube, such engagement involves online activities such as liking, disliking, commenting,

sharing, and uploading videos [22]. These features allow users to take a much more active role in the user experience, and thus they clearly suggest a deeper engagement with the content [23]. Furthermore, these popularity indicators are known to increase video views. For example, the number of comments and likes are significant predictors of video view counts on YouTube, resulting in the greater likelihood of marking a video as a favorite and choosing to rate a video [23,24]. The likes and comments posted on the videos spread by word of mouth, which potentially leads to more views [25]. Channel subscription is also an important measure of user engagement, as it reflects a sustained interest in the channel [26]. The number of comments, dislikes, likes, and the number of subscribers to a specific channel are classified as popularity-driven measures [27,28]. Apart from the popularity-driven measures, heuristic-driven measures such as the length of videos and the number of elapsed days after videos are uploaded are positively associated with the video view count [29]. Thus, we propose the following hypotheses.

**Hypothesis 1a (H1a).** *The number of comments will be positively related to the number of views.*

**Hypothesis 1b (H1b).** *The number of likes will be positively related to the number of views.*

**Hypothesis 1c (H1c).** *The number of subscribers will be positively related to the number of views.*

**Hypothesis 1d (H1d).** *The running time of content will be positively related to the number of views.*

Social media store the many activities and relationships formed in pertinent spaces in the form of data. Such stored data are nothing more than fragmental information when considered individually. However, if such information is examined macroscopically, the social networks formed on the basis of active interactions can be discovered [6]. A network refers to a structure in which the respective objects are connected with each other. If a social network is analyzed, the relationships between objects can be visualized using a graph, and social phenomenon or relational structure can be identified [30]. Therefore, if various tourism-related information uploaded on social media are collected and structuralized as a network, the flow of information and the shape of concentrated information can be identified comprehensively.

According to the perspective of content propagation in social network research [31,32], a great number of content users lead to a higher content propagation rate. When a video is shared on YouTube at a very high rate, it might then be classed as having viral popularity or being viral [33,34]. When video content is consumed by highly influential users, this content will become popular very easily by means of viral spread among the peers of these highly influential users. These viral videos can become very popular within a social network for a short period. Hence, the popularity of any given video is affected by the structural characteristics of the video viewers' social network. Supporting this suggestion, Yoganarasimhan [35] found that the size and structure of the video generators' network were significant determinants of the popularity of their videos.

In addition to user-to-user networks, other researchers have examined the structural connections between different content objects as an important factor in disseminating the content on YouTube [36]. On YouTube, videos are related to one another when they are topically similar. The connections between these videos can be explored through network analysis. Social network analysis (SNA) provides a sociological approach to the examination of relationships between content objects as well as the structure of social relationships and interactions between human beings [37]. In addition, SNA provides a variety of methodologies and indicators to examine node links and to present a structural pattern of connected systems [38]. However, despite the potential benefits of SNA, little research has examined what and how the structural factors of the content network are linked to the popularity of travel videos on YouTube. To investigate the potential of network

structure aspects as predictors of viewing YouTube travel videos, the following research questions are posited:

**RQ1a.** *What is the relationship between degree centrality and the number of views?*

**RQ1b.** *What is the relationship between betweenness centrality and the number of views?*

**RQ1c.** *What is the relationship between eigenvector centrality and the number of views?*

### 3. Methodology

*3.1. Data Collection and Preparation*

To collect the videos on Incheon Chinatown, we used the YouTube video network collection tool NodeXL. The search queries were performed in English so that both domestic and foreign tourists could search for them. As for the search terms, a preliminary investigation was conducted through Google Trends, which confirmed that "city name" was in the top search rankings with respect to the search words for Chinatown in each country around the world. Thus, "*Incheon Chinatown*" and "*Chinatown Incheon*" were used as the search queries and 500 videos were collected in May 2017. These videos were the maximum video clips allowed for collection by NodeXL. Table 1 shows the information regarding nodes and links for collected data. The nodes in the video networks were the videos related to Incheon Chinatown that were to be examined. Each collected video was manually examined to remove those irrelevant to Incheon Chinatown. As a result, 104 videos were included in the final analysis, and the number of links was reduced from 34,788 to 1463.

**Table 1.** Comparison of results before and after data refinement.

|  | Before Data Refinement | After Data Refinement |
|---|---|---|
| Number of nodes | 500 | 104 |
| Number of links | 34,788 | 1463 |

*3.2. Analysis Category*

The network that displays the relationships between the users of social media is represented as a graph, in which each node is connected with a link. Through this, it is possible to measure the relationship, closeness, and link strength between users [39]. The networks formed by the interactions between users are classified as non-directional or directional networks, and the network type is determined by the presence/non-presence of directionality in the link relationships between nodes. In this study, a video network analysis was conducted to examine how much interest YouTube users had on Incheon Chinatown, rather than to examine the relationships between YouTube video users. The video network is a non-directional network and the subscriber network is a directional network. Furthermore, in general network analysis, a user or subscriber is represented as a node, but in video networks, a node represents a video and a relationship between two nodes is represented based on tags that link the original video, reviews (comments) for the video, and reactions (likes) for the video [36]. In cases where there are more active reactions of video users, there are more links between the nodes and the distances are shorter.

To investigate the network structure of Incheon Chinatown videos that were searched for on YouTube, centrality analyses were performed using NodeXL. In this study, the centrality indices considered as analysis targets included degree centrality, betweenness centrality, and eigenvector centrality. Degree centrality is the number of direct connections a node has in a network [40]. A video with a high degree centrality score is regarded as a video with high information spreading power. Betweenness centrality measures how much a node acts as a bridge to all others in a network [40]. The video with a high betweenness centrality score is considered the information liaison in a network. Eigenvector centrality measures the importance of a node by the measure of its connectivity to other important



nodes in a network [41]. In addition to the centrality indices, we included the following engagement factors affecting the number of views and spreading of YouTube videos: (1) the number of comments that can measure user participation or empathy, (2) the number of likes that can identify user preferences, (3) the number of registered subscribers that can examine the reputation and awareness of content, and (4) the running time of content. As for each data point, if the criteria are selected when extracting data from NodeXL (developed by Marc Smith and his team at Microsoft Research), the result is automatically provided in which the running time of the content was calculated by converting the period between the first registration date and the extraction date of data to the number of days. Table 2 presents the descriptive statistics for all variables.

**Table 2.** Descriptive statistics for all variables.

|  | Min | Max | Mean | SD |
|---|---|---|---|---|
| Number of views | 1.00 | 40,478.00 | 1835.54 | 5245.696 |
| Number of comments | 0.00 | 181.00 | 11.22 | 31.51 |
| Number of likes | 0.00 | 1353.00 | 44.21 | 162.99 |
| Number of subscribers | 0.00 | 443,171.00 | 21,583.20 | 64,134.99 |
| Running time of content | 25.00 | 4033.00 | 873.21 | 882.51 |
| Degree centrality | 0.00 | 43.00 | 28.14 | 15.37 |
| Betweenness centrality | 0.00 | 3.78 | 0.49 | 1.10 |
| Eigenvector centrality | 0.00 | 0.02 | 0.01 | 0.01 |

## 4. Results

H1a–d predicted the positive associations between users' engagement factors and the number of views. In addition, RQ1a–c asked whether the structural factors of the content network were related to the number of views. To test the hypotheses and answer the research questions, we performed a multiple regression with the number of views as the dependent variable. Engagement and network structure factors were entered as predictors in the regression model. Before testing the regression model, screening revealed that some variables were not normally distributed. To address this, the variables were log-transformed, which is a standard practice used to normalize a skewed distribution of data. There was no violation of the absence of multicollinearity.

As shown in Table 3, the number of comments ($\beta = 0.21$, $p < 0.05$), the number of likes ($\beta = 0.57$, $p < 0.001$), and the running time of content ($\beta = 0.22$, $p < 0.01$) were positively related to the number of views. Specifically, for videos with higher numbers of comments, the higher numbers of likes, and the higher running times of content, indicated a greater number of views. However, the number of subscribers had no significant association with the number of views. Accordingly, H1a, H1b, and H1d were supported, but H1c was rejected.

**Table 3.** Regression analysis predicting the number of views.

| Predictors | $\beta$ | SE | t |
|---|---|---|---|
| Number of comments | 0.21 | 15.54 | 2.26 * |
| Number of likes | 0.57 | 4.05 | 4.65 *** |
| Number of subscribers | 0.12 | 0.01 | 1.43 |
| Running time of content | 0.22 | 0.29 | 3.32 ** |
| Degree centrality | −0.13 | 21.54 | −1.07 |
| Betweenness centrality | 0.02 | 263.10 | 0.27 |
| Eigenvector centrality | 0.07 | 388,335.10 | 0.63 |
| Adjusted $R^2$ | 0.75 | | |

*$p < 0.05$, ** $p < 0.01$, *** $p < 0.001$.

On the other hand, none of the three network structure factors (i.e., degree centrality, betweenness centrality, and eigenvector centrality) were significantly related to the number

of views. To better understand the non-significant associations between network structure factors and the number of views, we checked the network structure of 104 videos (see Figure 1). As shown in Figure 1, the nodes with high degree centrality are marked in red; the higher the betweenness of the node, the larger the size of the node is. Regarding the network structure of videos related to Incheon Chinatown, the density of each node was not high, and the links were shown to be broken depending on the group. Specifically, multiple red nodes were observed in two groups (groups containing the red dots) that had several links at each node. This indicates that each video was consumed at similar levels given that there was no representative video that could promote Incheon Chinatown. This result can be interpreted to suggest that when YouTube users are searching for Incheon Chinatown in English, the relevant videos are consumed fragmentarily and do not lead to the consumption of related videos. Furthermore, excluding the two groups that had relatively high link levels, the remaining nodes were either linked with a small number of nodes or existed only as single nodes, showing that related searches of Incheon Chinatown are not actively carried out.

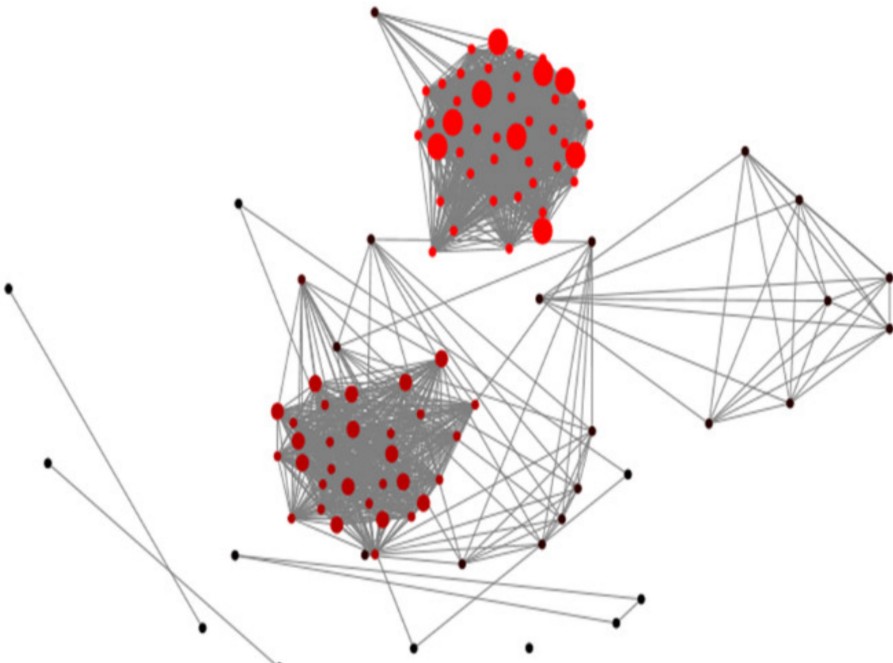

**Figure 1.** The network structure of YouTube videos about Incheon Chinatown.

## 5. Discussion

The current research improves the understanding of two factors predicting the popularity of tourism videos on YouTube: user engagement and video network structure. Consistent with previous studies on user engagement motivating video consumption on YouTube [22,23,42], the number of comments, the number of likes, and the running time of content were significant predictors of the viewing of travel videos on YouTube. These engagement indices can act as descriptive normative information, informed by perceptions of how many people watch, like, and post comments about a tourism video. The descriptive norms, in turn, can affect readers' behaviors and attitudes regarding the video.

According to the theory of normative social behavior [43], descriptive norms, that is, an individual's beliefs about the prevalence of a behavior, have a significant driving force to affect individuals' own behaviors. Social and user-generated sites such as YouTube enable users to learn about and deliver descriptive norms and social cues [44]. In particular, the virality of online content, as the number of likes, dislikes, and comments, can be recognized as an expression of descriptive social norms (e.g., "this is what most people do"). In the context of online communication, descriptive social norms can be observed through

algorithmic measures of message popularity and virality (e.g., high likes, shares, and comments) [45]. Thus, the popularity and virality might have contributed to high views.

Among four factors of user engagement, only the number of subscribers was found to be not associated with the number of views. This finding is in line with a recent study [25] that showed that the views of the YouTube COVID-19 videos were not dependent on the subscriber of the channel. At the most basic level, engagement begins with exposure, but the highest level of engagement is expression. The interactive nature of online media enables the audience to not only receive information but also create, exchange, and share it [46,47]. More specifically, interactive features available on YouTube, such as direct feedback, comment sections, and interactions with other users, allow users to take a much more active and interactive role for deeper engagement with YouTube content [23]. Compared to providing reaction or opinion regarding the YouTube tourism video, simply subscribing to a YouTube channel on tourism makes the audience less invested, aware, and attentive—namely, less engaged—with the content. For this reason, the subscription might have failed to elicit more views.

Another noteworthy finding is that the network structure factors (i.e., degree centrality, betweenness centrality, and eigenvector centrality) of video content did not predict the viewing of the video. One possible reason for this is that videos about Incheon tourism were unlikely to be organized by several powerful videos with central network positions in the YouTube network. In other words, there were no core videos, which involved not only the dissemination of information but also the bridge of other small communities in the network. This is different from previous results that popular videos rated by a number of viewers are highly clustered, disseminating similar ideas, and reinforcing solidarity among viewers [48].

These findings have practical implications for sustainable tourism marketing. First, the audience engagement elements, such as comments, likes, and running time, can be critical factors for YouTube tourism videos to be popular. Thus, it is recommended for the marketing manager and decision-maker to promote interactive engagement activities centered on the content as well as providing tourism content on YouTube. Second, it is necessary to create core videos that the greatest number of prospective customers can see on YouTube when considering Incheon Chinatown as a travel destination. The current video contents about Incheon Chinatown are watched in a fragmented way according to our findings. Core videos can be connected to other videos and can be recommended to other viewers after an individual has watched one video. This means that viewers continue to watch other videos after watching one video. As viewers watch more videos about Incheon Chinatown, it is more likely that they would then be more likely to consider traveling to Incheon Chinatown because visual stimulation is important when individuals decide where to travel [49].

Lastly, this study advances current research methods by using big data social media analytics in the field of YouTube research. Previous social media analytic studies on YouTube have focused on examining the relationships between YouTube users [50–52], but relatively little is known about object-object networks which explain the connections between YouTube videos about certain common topics. As a type of structural analysis, the social network analysis method is utilized to investigate characteristics of network structures, relationship natures of networks, positions of nodes embedded in networks, and communication patterns between nodes [53]. In addition, this study contributes to the development of sustainable tourism research. Although sustainable tourism research is comprehensive and multi-dimensional, many studies focus on climate change, behavioral studies, or tourism policies [54,55]. The current study extends methodological areas in sustainable tourism studies by highlighting big data analytics that can capture much larger information and potentially reflects longitudinal change in real-time [56].

Despite these implications, there are still limitations that need to be considered. First, this study focused on network structures of video content per se on YouTube. Future researchers could extend this study by analyzing the network structures in comment threads

on YouTube videos. Second, this study analyzed videos only available on YouTube. Given that other social media are considered as important destination marketing tools [57,58], future studies should explore the video contents on the different types of social media platforms. Third, this study did not consider the themes of videos. The volume of video views can vary across tourism themes, such as history, adventure, unusual geographic spots, fantasy, and futurism. Finally, future research should attempt to replicate these findings across various tourism themes.

In conclusion, YouTube has become the most effective platform for disseminating tourism content, which can foster sustainable tourism. In an effort to promote the discussion and sharing of tourism information and experience via YouTube videos, the current research suggests that active engagement and core content play a crucial role in predicting views for tourism content and managing sustainable tourism.

**Author Contributions:** Conceptualization, W.Y. and T.K.; Formal analysis, W.Y.; Funding acquisition, W.Y.; Investigation, S.L.; Methodology, T.K.; Writing—original draft, W.Y. and T.K.; Writing—review & editing, S.L. All authors have read and agreed to the published version of the manuscript.

**Funding:** This work was supported by Incheon National University (International Cooperative) Research Grant in 2019.

**Institutional Review Board Statement:** Not applicable.

**Informed Consent Statement:** Not applicable.

**Data Availability Statement:** Data available on request.

**Conflicts of Interest:** The authors declare no conflict of interest.

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
