# Peer review of "Predictors of Viewing YouTube Videos on Incheon Chinatown Tourism in South Korea: Engagement and Network Structure Factors"

_sustainability, doi:10.3390/su132212534_

Round 1

Reviewer 1 Report

I think this is a coherent and well-written article. It is succinct, links the findings with normative social behavior and the potential for user-generated social media content and sites to deliver descriptive norms and social cues. It also has applied research value for tourism destination marketers. From my perspective, this article is ready for publication. Congratulations on your work! 

Author Response

We thank the reviewers for their thoughtful and thorough feedback. We have taken every comment and critique seriously and addressed them to the best of our abilities. The result, we believe, is a much improved paper that integrates almost all of the suggestions offered by the reviewers. Please find attached our revised manuscript with the track changes. Below, we provide detailed responses to all of the substantive review comments.

Reviewer #1:

(1) Comment: I think this is a coherent and well-written article. It is succinct, links the findings with normative social behavior and the potential for user-generated social media content and sites to deliver descriptive norms and social cues. It also has applied research value for tourism destination marketers. From my perspective, this article is ready for publication. Congratulations on your work!

Response: We appreciate the reviewer’s thoughtful comment.

Reviewer 2 Report

Both the topic and the scientific approach are interesting, even if not innovative. The research hypotheses are not defined and no reference is made to them in the Discussion paragraph either.

Although they discuss the sustainability of tourism, the authors fail to make the connection between their own research and this theoretical and practical concept. It would be good to make the connection clearer in this regard, especially in the Conclusions paragraph.

Hypotheses (with reference to other studies), Discussions and Conclusions should be presented more clearly.

Author Response

We thank the reviewers for their thoughtful and thorough feedback. We have taken every comment and critique seriously and addressed them to the best of our abilities. The result, we believe, is a much improved paper that integrates almost all of the suggestions offered by the reviewers. Please find attached our revised manuscript with the track changes. Below, we provide detailed responses to all of the substantive review comments.

Reviewer #2:

(1) Comment: The research hypotheses are not defined and no reference is made to them in the Discussion paragraph either.

Response: As requested by the reviewer, we proposed new hypotheses (H1a-1d) with references (p. 4). We also the results of testing the hypotheses in Results section (p. 6). However, RQs were maintained because of limited empirical evidence on the relationship between the structural factors of the content network and the viewing of YouTube travel videos.

(2) Comment: Although they discuss the sustainability of tourism, the authors fail to make the connection between their own research and this theoretical and practical concept. It would be good to make the connection clearer in this regard, especially in the Conclusions paragraph.

Response: We appreciate constructive comment. As requested, we elaborated on the connection between the research findings and theoretical and practical concepts in the Discussion section (pp. 8-9).

(3) Hypotheses (with reference to other studies), Discussions and Conclusions should be presented more clearly.

Response: As requested, we added the hypotheses with reference (p. 4) and then revised the Discussion, including conclusions, more obviously (pp. 8-9). 

Reviewer 3 Report

The paper adopted an interesting research methodology and the author addressed the interesting issues of suing Youtube videos to investigate engagement and network structure. 

The author offered some background of Inchoen Chinatown as the site of the study. However, the significance of the site should be decribed more clearly, including the selection criteria among all other sites in Inchoen areas, for example. 

In addition, the research gap should be clearly discussed and presented to highlight the expected contribution of the current study. The author presented the research findings with the results of regression analysis. However, the author should also explain how the basic assumptions of regression analysis are met and the descriptive statistics of the all factors should also be provided. Furthermore, in the conclusions, the author should also explained the answers to all the RQ1a-d and RQ2a-c. The limitations of the study should also be mentioned. In the part of practical implications, since the contributors of Youtube videos are varied and uncontrollable, the authors should address the clearer guidelines and actions for the stakeholders to apply. Overall, the paper is interesting in the approach and can be a good guideline for future researchers. Thank you. 

Author Response

We thank the reviewers for their thoughtful and thorough feedback. We have taken every comment and critique seriously and addressed them to the best of our abilities. The result, we believe, is a much improved paper that integrates almost all of the suggestions offered by the reviewers. Please find attached our revised manuscript with the track changes. Below, we provide detailed responses to all of the substantive review comments.

Reviewer #3:

(1) Comment: The author offered some background of Incheon Chinatown as the site of the study. However, the significance of the site should be described more clearly, including the selection criteria among all other sites in Incheon areas, for example.

Response: We appreciate constructive comment. Thus, we described the significance of Incheon Chinatown in the Introduction section (p. 1) and heritage tourism contexts (p. 2).

(2) The research gap should be clearly discussed and presented to highlight the expected contribution of the current study.

Response: As requested, we added the research gap in the Introduction section (p. 2) and presented the expected contribution of this research in sustainable tourism studies (p. 8).

(3) The author should also explain how the basic assumptions of regression analysis are met and the descriptive statistics of the all factors should also be provided.

Response: As requested, we added the explanation how the basic assumptions were addressed or met before we ran a regression analysis (p. 7). We also provided the table (Table 2) that showed the descriptive statistics of all variables used in the analysis. (p. 6).  

(4) Furthermore, in the conclusions, the author should also explain the answers to all the RQ1a-d and RQ2a-c.

Response: As requested, we provided the answers to all of the research questions (RQ2a-2c) in the Discussion section (pp. 8-9). According to other reviewers’ opinions, we changed some research questions (RQ1a-d) to the hypotheses (H1a-d), and thus revised the Results (p. 6) and Discussion sections (pp. 8-9).

(5) The limitations of the study should also be mentioned. In the part of practical implications, since the contributors of YouTube videos are varied and uncontrollable, the authors should address the clearer guidelines and actions for the stakeholders to apply.

Response: Based on the findings of the current research, we added a clearer guideline for the stakeholder to apply in sustain tourism marketing (p. 8).

Reviewer 4 Report

The paper entitled “Predictors of Viewing YouTube Videos on Incheon Chinatown Tourism in South Korea: Engagement and Network Structure Factors” is, in my opinion, a solid piece of scientific work. The abstract brings all necessary information. I believe that the authors should make these revisions:

  1. The introductory part clearly explains the originality (research gap) of the paper and the paper's aim. However, the structure that is to carry out the development of the article is not defined
  2. The theoretical part is based on credible literary sources. Nevertheless, hypotheses development needs more strong theoretical arguments
  3. Regarding methodology, the study is well conducted and appropriate.
  4. The results are clearly formulated and are supported by appropriate tables and pictures. The scientific hypotheses were evaluated appropriately. However, a section explaining the main findings has not been incorporated. This is necessary so that the reader is aware of the main achievements of this article.

Author Response

We thank the reviewers for their thoughtful and thorough feedback. We have taken every comment and critique seriously and addressed them to the best of our abilities. The result, we believe, is a much improved paper that integrates almost all of the suggestions offered by the reviewers. Please find attached our revised manuscript with the track changes. Below, we provide detailed responses to all of the substantive review comments.

Reviewer #4:

(1) Comment: The introductory part clearly explains the originality (research gap) of the paper and the paper’s aim. However, the structure that is to carry out the development of the article is not defined

Response: As requested, we explained the structure how this research was carried out in the Introduction section (p. 2).

(2) Comment: The theoretical part is based on credible literary sources. Nevertheless, hypotheses development needs more strong theoretical arguments.

Response: As requested by the reviewer, we changed the research questions (RQ1a-1d) to the hypotheses (H1a-1d) based on theoretical frameworks and practical evidence (p. 4).

(3) Comment: The scientific hypotheses were evaluated appropriately. However, a section explaining the main findings has not been incorporated. This is necessary so that the reader is aware of the main achievements of this article.

Response: We appreciate the reviewer’s thoughtful comment. To highlight the main achievement of this research, we revised the Discussion section (pp. 8-9).